# Different Oncologic Outcomes in Early-Onset and Late-Onset Sporadic Colorectal Cancer: A Regression Analysis on 2073 Patients

**DOI:** 10.3390/cancers14246239

**Published:** 2022-12-18

**Authors:** Caterina Foppa, Annalisa Maroli, Sara Lauricella, Antonio Luberto, Carlotta La Raja, Francesca Bunino, Michele Carvello, Matteo Sacchi, Francesca De Lucia, Giuseppe Clerico, Marco Montorsi, Antonino Spinelli

**Affiliations:** 1Department of Biomedical Sciences, Humanitas University, Via Rita Levi Montalcini 4, Pieve Emanuele, 20090 Milan, Italy; 2IRCCS Humanitas Research Hospital, Division of Colon and Rectal Surgery, Via Manzoni 56, Rozzano, 20089 Milan, Italy

**Keywords:** early-onset, colorectal cancer, late-onset, oncologic outcomes, age of onset, progression free survival, cancer specific survival

## Abstract

**Simple Summary:**

Early-onset colorectal cancer (EOCRC) is on the rise. Reasons are unknown and comparative results about long-term survival are still widely debated. This study aimed to explore the effect of early age of onset in a surgical population of sporadic colorectal cancer patients. Early age of onset resulted independently and is associated with worse oncologic outcomes also in stage I patients. This study suggested that EOCRC may have a more aggressive tumoral phenotype compared with late-onset CRC. A better understanding of the biology of EOCRC is needed to revise —and eventually modify—current strategies of treatment and surveillance.

**Abstract:**

The incidence of colorectal cancer (CRC) is increasing in the population aged ≤ 49 (early-onset CRC-EOCRC). Recent studies highlighted the biological and clinical differences between EOCRC and late-onset CRC (LOCRC-age ≥ 50), while comparative results about long-term survival are still debated. This study aimed to investigate whether age of onset may impact on oncologic outcomes in a surgical population of sporadic CRC patients. Patients operated on for sporadic CRC from January 2010 to January 2022 were allocated to the EOCRC and LOCRC groups. The primary endpoint was the recurrence/progression-free survival (R/PFS). A total of 423 EOCRC and 1650 LOCRC was included. EOCRC had a worse R/PFS (*p* < 0.0001) and cancer specific survival (*p* < 0.0001) compared with LOCRC. At Cox regression analysis, age of onset, tumoral stage, signet ring cells, extramural/lymphovascular/perineural veins invasion, and neoadjuvant therapy were independent risk factors for R/P. The analysis by tumoral stage showed an increased incidence of recurrence in stage I EOCRC (*p* = 0.014), and early age of onset was an independent predictor for recurrence (*p* = 0.035). Early age of onset was an independent predictor for worse prognosis, this effect was stronger in stage I patients suggesting a potentially—and still unknown—more aggressive tumoral phenotype in EOCRC.

## 1. Introduction

The incidence of early-onset colorectal cancer (EOCRC-age of onset ≤ 49 years) has progressively risen worldwide [1,2,3,4]—approximately 2% per year since 1994—while CRC incidence rates in people aged ≥ 50 (late-onset CRC—LOCRC) have remained stable or declined in many countries thanks to screening programs. A recent systematic review [5] of 40 studies crossing 12 countries and five continents has reported a worldwide 30% increase in the incidence of EOCRC over the past 20 years. The reasons underlying this rise are poorly understood and not explained by hereditary syndromes as the increased incidence is seen among sporadic cases, with considerable genotypic and phenotypic heterogeneity [6,7]. Some of the hypothesized factors (obesity, smoking, sedentary behavior, and unhealthy diet) cannot completely explain this phenomenon, as they are not specifically age-related. According to the current literature, EOCRC are more located in the left colon and rectum, have often a mucinous and signet ring cells histology, a poorer tumor differentiation, and are more advanced at diagnosis when compared to LOCRC [1,4]. Comparative results about long-term survival are still discordant [8,9,10,11,12]. The lack of data and standardization of the cut-off age may explain the discrepancies and the impossibility to interpret the reported outcomes. A better understanding of the characteristics and etiology of EOCRC—and its differences with the late-onset counterpart—may help to promote effective prevention, early detection, and treatment strategies. A comparative study between sporadic early- and late-onset rectal cancer (RC) reporting age of onset as an independent predictor for disease progression and recurrence and worse oncologic outcomes in stage I patients was recently published by our group [12]. We here aimed to investigate whether the age of onset may impact on disease recurrence/progression in sporadic colon and rectal cancers analyzed together.

The contribution/significance of this article:It gives a contribution to the still-debated oncologic outcomes in EOCRC.It is the first report that early age of onset is a strong predictor for worse oncologic outcomes in CRC patients.It is the first report that EOCRC have a worse prognosis at earlier stages,It gives a contribution to the still limited comparative data on a large surgical population of sporadic early- and late-onset CRC.The results may be a starting point for further research on the topic.

## 2. Materials and Methods

### 2.1. Study Design and Patients’ Selection

This was a tertiary referral center ambidirectional parallel-cohort study. The study was approved by the Independent Ethical Committee of the Institution. Data on patients operated on for CRC from 1 January 2010 to 31 January 2022 were retrospectively collected from the prospective database of our division. A further review of variables was performed when needed. The list of all prospective variables of the database is detailed in Appendix A. All consecutive sporadic colorectal adenocarcinomas operated on in the time frame were included. Patients 50 years old or older were allocated to the LOCRC group; patients aged 49 or younger were allocated to the EOCRC cohort. Exclusion criteria included palliative surgery (e.g., colostomy or ileostomy construction), local excision (by endoscopy or surgery), surgical indication for benign lesions (i.e., adenomatous polyps not endoscopically removable), diagnosis of anal spinocellular cancer, histological diagnosis different from colorectal adenocarcinoma, concomitant diagnosis of inflammatory bowel diseases, known genetic syndromes, and a significant proportion of missing data (with a threshold of 5%) (Appendix A).

### 2.2. Endpoints and Variables

The primary endpoint of the study was to compare the incidence rate ratio of CRC recurrence or progression between LOCRC and EOCRC cohorts. Recurrence consisted of any (local or systemic) evidence of disease after surgery. Patients presenting with metastatic disease at diagnosis and reporting an increased metastasis size or number (in the same or other organs) were defined as experiencing a disease progression. Disease persistence was defined as stable disease (number and size of metastasis) during the follow-up in a stage IV patient at diagnosis; these patients were censored in the analysis.

The detailed list of collected variables is reported in Appendix A: briefly, all relevant demographic, clinical, therapeutics, radiological, surgical, pathological, and oncological data were selected from the prospective database to be analyzed in the study.

Right colon cancer (RCC) was defined as cancer from the ileocecal valve to the mid-transversum; left colon cancer (LCC) was defined as cancer from the mid-transversum to the rectosigmoid junction; rectal cancer (RC) was defined by magnetic resonance imaging (MRI) as cancer located below 15 cm from the ano-cutaneous verge. Pathological staging was performed following the American Joint Committee on Cancer (AJCC) TNM classification [13]. Stage IV cases at diagnosis and RC cases were discussed multidisciplinary at the tumor board according to our daily clinical practice.

A positive family history was registered in the case of one or more first/second degree relatives affected by CRC for patients without Amsterdam or Bethesda criteria for hereditary non-polyposis colon cancer (HNPCC) syndrome or clinical criteria for familial adenomatous polyposis (FAP) [14]. Detection of deleterious mutations in DNA mismatch repair genes identified HNPCC (or Lynch syndrome) and represented an exclusion criterion from the present analysis. LOCRC patients were genetically tested only if in case of strong family history, proven genetic syndromes in the family, history of other primary tumors or microsatellite instability (MSI), which was tested for all patients since 2009.

### 2.3. Statistical Analysis

Categorical and binomial variables are reported as percentages over the total. Continuous variables were tested for normality with the Shapiro–Wilk test and are reported as mean ± standard deviation or median and interquartile range [IQR]. Missing data were analyzed for pattern distribution and imputed using a regression-based multiple imputation model. Categorical and dichotomous variables were analyzed using Pearson’s Chi-test with Fisher’s exact test. Continuous data were compared with T-test (normal variables) or a Mann–Whitney test (non-normal variables). All tests were unpaired and two-sided with an α-level of 0.05. Recurrence and progression (R/P) and cancer-specific (CS) incidence rates were analyzed by the Kaplan–Meier curves with log-rank (Mantel–Cox) test. The cumulative effect of multiple variables on progression and recurrence will be analyzed with a Cox proportional hazards regression model. For each variable, the hazard ratios (HR) and 95% confidence intervals (95% CI) were reported. Dead or lost-to-follow-up patients were censored at the time of death or last follow-up. Analyses were performed using IBM SPSS Statistics for Windows, Version 24.0 (Armonk, NY, USA: IBM Corp.). Graphics were made with GraphPad Prism 5 Software (GraphPad Software, Inc., San Diego, CA, USA).

## 3. Results

Out of 2578 patients undergoing colorectal resection in the study period, 2073 were included (423 EOCRC, 1650 LOCRC).

### 3.1. Demographics and Clinical Presentation

Mean age at diagnosis was 68.78 (±10.07) in LOCRC and 42.44 (±5.90) in EOCRC. Female gender was more represented in EOCRC (50% vs. 41%; *p* = 0.003). Smoking habit was more frequently reported in EOCRC (23% vs. 16%; *p* < 0.0001). EOCRC were more frequently located in the left colon (22% vs. 32%; *p* < 0.0001) and metastatic at diagnosis (34% vs. 15%; *p* < 0.0001). Accordingly, the EOCRC cohort presented a higher percentage of patients undergoing preoperative chemotherapy and/or radiotherapy compared with LOCRC (49% vs. 35%; *p* < 0.0001). CRC family history was more frequently reported by EOCRC patients compared with LOCRC (25% vs. 20%; *p* = 0.048). The presence of comorbidities was higher in the LOCRC group (*p* < 0.0001). Body mass index (BMI) (*p* = 0.055), proportion of obesity (*p* = 0.528), synchronous tumors (*p* = 0.164), and family history of cancer (*p* = 0.258) did not differ (Table 1).

### 3.2. Operative Data and Postoperative Outcomes

A significantly higher proportion of metastatic LOCRC patients underwent surgery for either CRC and metastases (in one or two stages) (94% vs. 90%; *p* = 0.003).

A higher proportion of EOCRC underwent surgery in an emergency setting (4% vs. 1%; *p* = 0.002) and was operated on by an open approach (28% vs. 18%; *p* < 0.0001). Operative time did not differ in the two groups (*p* = 0.114). The rate of postoperative complications was higher in the LOCRC (29% vs. 21.5%; *p* = 0.010); however, the length of hospital stay was longer in the EOCRC group (*p* < 0.0001). No differences in the reoperation rate were reported (7% in EOCRC vs. 8% in LOCRC; *p* = 0.892) (Table 2).

### 3.3. Pathological and Molecular Features

A lower proportion of EORC patients had a major or complete response to neoadjuvant radio/chemotherapy according to the Dworak scale (37% vs. 51%; *p* = 0.005). A higher rate of nodal (52% vs. 38.75%; *p* < 0.0001) and distant metastases (34% vs. 15%; *p* < 0.0001) was observed in the EOCRC group. The number of harvested lymph nodes was higher in the EOCRC group (*p* = 0.0002). The proportion of mucinous, signet ring cells, and microsatellite instable tumors were comparable among the cohorts (*p* = 0.513, *p* = 0.169, and *p* = 0.113, respectively). The occurrence of KRAS/BRAF/NRAS/Pi3Kca mutations was higher in EOCRC (51% vs. 13%; *p* < 0.0001). In a higher number of EOCRC patients the tumor grading could not be assessed (*p* < 0.0001). The rate of R0 resections did not differ (*p* = 0.09) (Table 3).

### 3.4. Oncological Outcomes

More patients in the EOCRC group underwent adjuvant therapy (73% vs. 48%; *p* < 0.0001). The rate of metachronous colon cancer was comparable (*p* = 0.073), while more LOCRC patients developed other neoplasms during the follow-up period (19% vs. 4%; *p* < 0.0001) (Table 4).

At a median follow-up of 28 [9–61] and 27 [7–60] months for EOCRC and LOCRC, respectively (*p* < 0.0001), a significantly higher incidence rate of R/P was observed in EOCRC patients compared with LOCRC (HR = 1.89; 95% CI: 1.72–2.75; *p* < 0.0001) (Figure 1).

The R/PFS proportion in EOCRC patients at 36 months was 51% versus 71% in the LOCRC group. The median survival of the EOCRC group was 43 months. EOCRC patients also showed a worse CSS (Appendix A) (HR = 2.09; 95% CI: 1.77–3.46; *p* < 0.0001).

### 3.5. Univariable and Multivariable Analysis

A Cox regression analysis was performed to explore the possible risk factors associated with a worse R/PFS in the study population (Table 5). At univariable analysis, the predictors for R/PFS were early age of onset (HR = 1.89; 95% CI: 1.72–2.75; *p* < 0.0001), neoadjuvant therapy (HR = 1.58; 95% CI: 1.31–1.90; *p* < 0.0001), advanced pathology stage (*p* < 0.0001), mucinous (HR = 1.39; 95% CI: 1.11–1.96; *p* = 0.005) and signet ring cell (HR = 1.58; 95% CI: 1.48–4.47; *p* < 0.0001) histology, extramural invasion (HR = 2.64; 95% CI: 2.19–3.18; *p* < 0.0001), lymphovascular invasion (HR = 2.68; 95% CI: 2.24–3.21; *p* < 0.0001), and perineural invasion (HR = 3.29; 95% CI: 2.66–4.03; *p* < 0.0001). Combined surgery on primitive CRC and metastasis (HR = 0.21; 95% CI: 0.16–0.27; *p* < 0.0001) resulted as a protective factor for recurrence or progression. At multivariable analysis, early age of onset (HR = 1.35; 95% CI: 1.09–1.67; *p* = 0.006), preoperative radio/chemotherapy (HR = 1.34; 95% CI: 1.09–1.63; *p* = 0.005), advanced tumoral stage (*p* < 0.0001), signet ring cells phenotype (HR = 2.24; 95% CI: 1.26–3.98; *p* = 0.006), extramural invasion (HR = 1.41; 95% CI: 1.14–1.74; *p* = 0.001), lymphovascular invasion (HR = 1.31; 95% CI: 1.06–1.62; *p* = 0.013), and perineural invasion (HR = 1.44; 95% CI: 1.14–1.81; *p* = 0.002) were confirmed as independent risk factors for recurrence or progression.

### 3.6. Stage-Dependent R/PFS Analysis

As R/PFS was strongly influenced by the tumoral stage, a sub-group analysis was performed to ascertain the effect of age of onset on the risk of developing recurrence or progression in each tumoral stage. Early age of onset was significantly associated with worse RFS (HR = 2.68; 95% CI: 1.07–6.72; *p* = 0.035) in stage I. The RFS proportion of EOCRC at 36 months was 81% versus 91% in the LOCRC group (Figure 2A). No differences were found between EOCRC and LOCRC in stage II (HR = 0.98; 95% CI: 0.53–1.83; *p* = 0.955), III (HR = 1.43; 95% CI: 0.92–2.20; *p* = 0.108), and IV (HR = 1.25; 95% CI: 0.93–1.67; *p* = 0.103) (Figure 2B–D). In stage II, the RFS proportion of EOCRC patients at 36 months was 74% versus 81%; in stage III, the RFS proportion of EOCRC patients at 36 months was 55% versus 63% in the LOCRC group; and in stage IV, the R/PFS proportion of EOCRC patients at 36 months was 18% versus 24% in the LOCRC group. The median survival of the LOCRC and EOCRC stage IV was 13 months and 11 months, respectively.

At multivariable analysis, early age of onset resulted as an independent risk factor for worse RFS in stage I (HR: 2.49; 95% CI: 1.19–5.23; *p* = 0.016) (Table 6).

## 4. Discussion

This study aimed to investigate the effect of age of onset on R/PFS in a large cohort of sporadic EOCRC and LOCRC patients operated on for the primary tumor at a single-tertiary center. Early age of onset resulted as a risk factor for progression/recurrence. Interestingly, at a stage-dependent analysis, early age of onset was significantly associated with worse RFS in stage I patients.

Consistently with this result, we previously found a poorer RFS in stages I–II adolescents and young adults (AYA) RC patients when compared to a matched late-onset cohort [11]. Additionally, we reported a worse RFS in stage I EORC patients when compared to the LORC counterpart [12]. Age > 65 years was identified as an independent factor associated with a lower risk for nodal metastases in T1 CRC in two recently published studies [15,16]. Although the cut-off age does not correspond to EOCRC, this result indicates that in early-stage CRC there is an oncological risk related to a younger age, which should be further investigated.

We did not report any difference in R/PFS at other stages while the rate of cancer-specific survival (CSS) was lower in the EOCRC group. Results on disease-free survival (DFS) and CSS in EOCRC are contrasting and still a matter of controversy [17]. In fact, some studies report worse [10,11,12] while others equivalent or better oncological outcomes among younger patients [17,18,19]. This can reflect as well different healthcare policies between nations. A study [20] reported a worse DFS in younger patients when they were stratified according to age at diagnosis: ≤40 vs. 41–64 vs. ≥65 and 21–30 vs. 31–40 vs. 41–50. The DFS did not change when they compared two groups of patients with the cut-off age of 50. Possible explanations for the reported discrepancies in survival outcomes among studies may be a lack of data and standardization—as different cut-off ages are set—preventing a homogeneous scenario for the interpretation of data.

Our results may suggest a more aggressive behavior of EOCRC, maybe due to still unknown biological features. The molecular underpinnings of EOCRC should be further investigated and—according to results—management, treatment, and surveillance should be revised. For example, the deficiency of the receptor for complement C3 anaphylatoxin C3a (C3aR) and the consequent increased expression of PV1 was described to promote the development of metastases particularly when tumors are left-sided [21]. This process closely resembles the mechanisms of EOCRC development and is worth further investigation in this subgroup of CRC patients. Recently, Marx et al. reported deregulation of the proto-oncogene MYC as the driver for the development of EOCRC and suggested a possible sub-classification of EOCRC based on MYC expression [22]. This was the first report identifying MYC as an oncogene in EOCRC. Point mutations in gene(s) involved in keeping correct chromosome segregation have been also hypothesized to be involved in EOCRC pathogenesis. Biological, molecular, and genetic features of EOCRC should be further investigated.

Recent studies suggest that EOCRC are different from LOCRC on both a biological and pathological standpoint [23,24], this may consequently have an impact on the response to multimodal treatment [24]. However, data are scarce and seem not to explain the worse prognosis of EOCRC. In our study, signet ring cells phenotype, lymphovascular, and perineural and extramural veins invasion were the pathological predictor for worse R/PFS. The only feature among these, which significantly differed between EO- and LOCRC, was lymphovascular invasion, with more reported in LOCRC.

Neoadjuvant treatment resulted as an independent predictor for worse R/PFS. More EORC patients presented with a locally advanced disease, requiring a neoadjuvant treatment. However, the response to treatment—according to Dworak pathology stage—was worse when compared to LORC. A study on 43,106 RC analyzed from a US cancer database demonstrated that stage II and III EORC patients undergoing treatment according to National Comprehensive Cancer Network guidelines did not have the expected survival benefits [10]. This result may support the hypothesis that tumors in young patients have different characteristics (biological, molecular, genetic), which in turn may impact the response to current therapeutic regimens. Future research should focus on confirming this hypothesis and, accordingly, to identify new treatment regimens for a targeted therapy. In fact, the impact of chemo-radiotherapy on quality of life (social, working, sexual sphere) should be particularly kept in mind in young patients where treatment success and oncologic outcomes should be well weighted.

From a surgical standpoint a higher rate of EOCRC underwent surgery by laparotomy. This data can be explained by several factors: the more advanced disease at diagnosis, the higher rate of surgeries conducted in an emergency setting and the higher rate of simultaneous interventions on the primary tumor and metastases, reflecting an inclination in being more surgically “aggressive” in young patients. These factors may also explain the longer length of stay in EOCRC, despite the lower complication rate reported in this cohort.

The limitations of this study are inherent to its partially retrospective design and the wide time spanning, as therapies have changed. The monocentric nature can be both considered as a limitation and as one of the strengths, because the single institutional database allowed for a greater accuracy in data gathering and analysis. Another limitation relies on the high disproportion among the two samples, possibly leading to a random bias. This study has also some strong points: the very strict age-definition of EOCRC and the large surgical sample of sporadic EOCRC patients offers a significant amount of data. This is the first work reporting early age of onset as an independent risk factor for worse prognosis, particularly at early stages. Furthermore, our work gives a contribution to the still-debated topic of oncologic outcomes in EOCRC.

## 5. Conclusions

In conclusion, early age of onset resulted in a risk factor for disease recurrence even at stage I. This result might reflect different and not yet defined biological, molecular, or genetic characteristics in EOCRC. The findings of our study warrant, in our opinion, a multicentric trial using the same strict inclusion criteria to confirm our results on a larger scale and to deepen the knowledge on the biologic underpinnings of EOCRC.

## Figures and Tables

**Figure 1 cancers-14-06239-f001:**
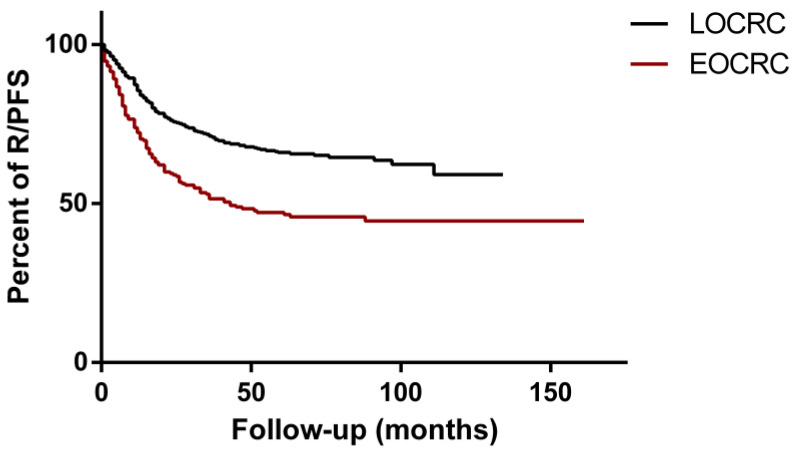
R/PFS (recurrence/progression-free survival) of EOCRC (early-onset colorectal cancer) (red line) and LOCRC (late-onset colorectal cancer) (black line) patients. Data were compared with Kaplan–Meier analysis and log-rank (Mantel–Cox) test (HR = 1.89; 95% CI: 1.72–2.75; *p* < 0.0001).

**Figure 2 cancers-14-06239-f002:**
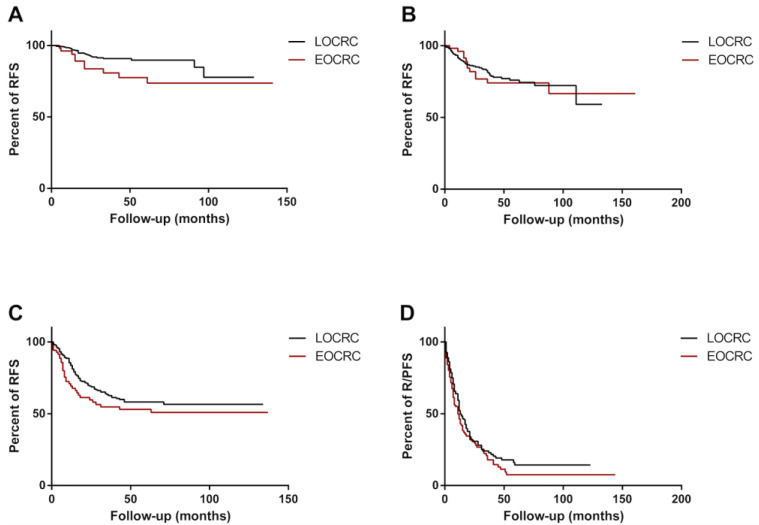
R/PFS (recurrence/progression-free survival) of EOCRC (early-onset colorectal cancer) and LOCRC (late onset colorectal cancer) patients according to the postoperative tumoral stage: (**A**) RFS of EOCRC (red line) and LOCRC (black line) patients diagnosed with tumoral stage I were compared with Kaplan–Meier analysis and log-rank (Mantel–Cox) test (HR = 2.68; 95% CI: 1.07–6.72; *p* = 0.035). (**B**) RFS of EOCRC (red line) and LOCRC (black line) patients diagnosed with tumoral stage II were compared with Kaplan–Meier analysis and log-rank (Mantel˜Cox) test (HR = 0.98; 95% CI: 0.53–1.83; *p* = 0.955). (**C**) RFS of EOCRC (red line) and LOCRC (black line) patients diagnosed with tumoral stage III were compared with Kaplan˜Meier analysis and log-rank (Mantel˜Cox) test (HR = 1.43; 95% CI: 0.92–2.20; *p* = 0.108). (**D**) R/PFS of EOCRC (red line) and LOCRC (black line) patients diagnosed with tumoral stage IV were compared with Kaplan–Meier analysis and log-rank (Mantel–Cox) test (HR = 1.25; 95% CI: 0.93–1.67; *p* = 0.103).

**Table 1 cancers-14-06239-t001:** Baseline and clinical characteristics, mean ± standard deviation, median [IQR], and % (n).

Characteristic	LOCRC	EOCRC	*p*
Number of patients	1650	423	
Age, years	68.78 ± 10.07	42.44 ± 5.90	<0.0001
Gender, females	41% (676)	50% (212)	0.003
BMI, Kg/m^2^	25.21 ± 4.13	24.73 ± 4.40	0.055
Obesity (BMI ≥ 30)	17% (280)	16% (68)	0.528
Smoking status			<0.0001
Non-smokers	60% (990)	65% (275)	
Current smokers	16% (264)	23% (97)	
Ex-smokers	24% (396)	12% (51)	
Tumor location			<0.0001
Right colon	27% (445)	22% (93)	
Left colon	22% (363)	32% (135)	
Rectum	51% (842)	46% (195)	
Relevant comorbidities	80% (1320)	21% (89)	<0.0001
Preoperative treatment	35% 585	49% 207	<0.0001
RT-CHT _(RC patients)_	44% (375)	51% (100)	
RT-CHT + systemic CHT _(RC patients)_	6% (54)	16% (32)	
Systemic CHT _(RC or CRC patients)_	9% (156)	18% (75)	
Synchronous tumors	3% (50)	2% (8)	0.164
Right colon	38% (19)	17% (1)	
Left colon	43% (22)	83% (7)	
Rectum	19% (10)	--	
Metastatic disease	15% (248)	34% (144)	<0.0001
Family history of cancer	48% (792)	44% (186)	0.285
Family history of CRC	20% (330)	25% (106)	0.048

Abbreviations: IQR, interquartile range; LOCRC, late-onset colorectal cancer; EOCRC, early-onset colorectal cancer; BMI, body mass index; CRC, colorectal cancer. RC, rectal cancer; RT, radiotherapy; CHT, chemotherapy. Continuous data are analyzed with unpaired two-sided T- or Mann–Whitney tests; dichotomous and categorical data are analyzed with Pearson’s χ^2^-test and Fisher’s exact test.

**Table 2 cancers-14-06239-t002:** Operative data and postoperative outcomes, mean ± standard deviation, median [IQR], and % (n).

Outcome	LOCRC	EOCRC	*p*
Number of patients	1650	423	
Combined surgery on CRC and metastases (one or two stage)	94% (233)	90% (130)	0.003
Surgical setting			0.002
Elective surgery	99% (1634)	96% (406)	
Emergent surgery	1% (16)	4% (17)	
Surgical approach			<0.0001
Minimally invasive surgery	82% (1353)	72% (305)	
Open surgery	18% (297)	28% (118)	
Operative time, minutes	230.88 ± 93.23	239.92 ± 110.17	0.114
Length of hospital stay, days	5 [3–7]	5 [5–8]	<0.0001
Ninety-day postoperative complications	29% (478)	22% (93)	0.010
Clavien-Dindo I	23% (110)	12% (11)	
Clavien-Dindo II	38% (182)	43% (40)	
Clavien-Dindo IIIa	11% (52)	16% (15)	
Clavien-Dindo IIIb	22% (105)	27% (25)	
Clavien-Dindo IVa	5% (23)	1% (1)	
Clavien-Dindo IVb	1% (6)	1% (1)	
Clavien-Dindo V	--	--	
Ninety-day postoperative reoperations	8% (132)	7% (29)	0.892

Abbreviations: IQR, interquartile range; LOCRC, late-onset colorectal cancer; EOCRC, early-onset colorectal cancer; CRC, colorectal cancer. Continuous data are analyzed with unpaired two-sided T- or Mann–Whitney tests; dichotomous and categorical data are analyzed with Pearson’s χ^2^-test and Fisher’s exact test.

**Table 3 cancers-14-06239-t003:** Pathological and molecular features, median [IQR], and % (n).

Feature	LOCRC	EOCRC	*p*
Number of patients	1650	423	
Tumor regression (Dworak classification)			0.019
Dworak 0	4% (17)	11% (14)	
Dworak 1	20% (86)	23% (31)	
Dworak 2	24% (103)	29% (38)	
Dworak 3	35% (150)	25% (33)	
Dworak 4	17% (73)	12% (16)	
Major or complete response to neoadjuvant therapy	24% (103)	34% (45)	0.005
Tumoral stage, AJCC 8th edition			<0.0001
Stage 0	5% (70)	5% (16)	
Stage I	26% (428)	18% (61)	
Stage II	29% (476)	19% (65)	
Stage III	25% (414)	25% (86)	
Stage IV	15% (248)	33% (116)	
Tumoral stage			0.064
0	5% (82)	6% (20)	
1	14% (215)	11% (39)	
2	19% (319)	14% (50)	
3	50% (818)	52% (178)	
4	12% (202)	17% (57)	
Node stage			<0.0001
0	61% (1002)	48% (165)	
1c	2% (38)	2% (6)	
1	24% (385)	25% (87)	
2	13% (211)	25% (86)	
Metastasis stage			<0.0001
0	85% (1388)	66% (228)	
1	15% (248)	34% (116)	
Number of harvested lymph nodes	22 [17–29]	23 [19–34]	0.0002
Lymph nodes ratio	0.10 [0.05–0.22]	0.29 [0.07–1.00]	<0.0001
Mucinous tumor	15% (249)	17% (57)	0.513
Signet-ring cells phenotype	1% (25)	3% (9)	0.169
Extramural invasion	27% (441)	22% (76)	0.068
Lymphovascular invasion	33% (594)	27% (93)	0.026
Perineural invasion	12% (197)	16% (54)	0.074
Tumor differentiation			<0.0001
Grade 0	1% (18)	1% (1)	
Grade 1	2% (30)	1% (6)	
Grade 2	44% (726)	38% (131)	
Grade 3	36% (586)	28% (98)	
Grade 4	1% (17)	1% (1)	
Not determinable	16% (256)	31% (107)	
Resection margins			0.090
R0	98% (1604)	99% (343)	
R1	1.9% (31)	1% (1)	
R2	0.1% (1)	--	
Microsatellite Instability	6% (99)	4% (17)	0.113
Mutations ‡	13% (203)	51% (78)	<0.0001

Abbreviations: IQR, interquartile range; LOCRC, late-onset colorectal cancer; EOCRC, early-onset colorectal cancer; BMI, body mass index; AJCC, American Joint Committee on Cancer. Continuous data are analyzed with unpaired two-sided T- or Mann–Whitney tests; dichotomous and categorical data are analyzed with Pearson’s χ^2^-test and Fisher’s exact test; ‡ On 1707 (LOCRC = 1554; EOCRC = 153) analyzed patients.

**Table 4 cancers-14-06239-t004:** Oncological outcomes, median [IQR], and % (n).

Outcomes	LOCRC	EOCRC	*p*
Number of patients	1650	423	
Metachronous CRC tumors	2% (33)	0.2% (1)	0.073
Other neoplasms	19% (314)	4% (55)	<0.0001
Adjuvant therapy	48% (792)	73% (309)	<0.0001
Follow-up, months	27 [7–60]	28 [9–61]	0.948
Death	12% (198)	22% (93)	<0.0001
Cancer-specific death	8% (132)	22% (93)	<0.0001
Recurrence or progression	21% (346)	40% (169)	<0.0001

Abbreviations: IQR, interquartile range; LOCRC, late-onset colorectal cancer; EOCRC, early-onset colorectal cancer; CRC, colorectal cancer. Continuous data are analyzed with unpaired two-sided T- or Mann–Whitney tests; dichotomous and categorical data are analyzed with Pearson’s χ^2^-test and Fisher’s exact test.

**Table 5 cancers-14-06239-t005:** Cox proportional hazard regression analysis on R/PFS.

	Univariable Analysis			Multivariable Analysis		
Variable	HR	95% CI	*p*	HR	95% CI	*p*
Age of onset (vs. LOCRC)	1.89	1.72–2.75	<0.0001	1.35	1.09–1.67	0.006
Gender (vs. males)	0.89	0.74–1.07	0.218	--	--	--
Obesity (BMI ≥ 30 Kg/m^2^)	1.07	0.85–1.36	0.545	--	--	--
Smoking (vs. non-smokers)			0.505	--	--	--
Current smokers	1.04	0.81–1.33	0.779	--	--	--
Ex-smokers	1.14	0.92–1.41	0.242	--	--	--
Tumor location (vs. right colon)			0.146	--	--	--
Left colon	1.10	0.86–1.41	0.448	--	--	--
Rectum	0.89	0.71–1.11	0.294	--	--	--
Synchronous tumors	1.40	0.87–2.24	0.163	--	--	--
Family history of CRC	0.84	0.66–1.06	0.134	--	--	--
Preoperative radio/chemotherapy	1.58	1.31–1.90	<0.0001	1.34	1.09–1.63	0.005
Surgery on CRC and metastases	0.21	0.16–0.27	<0.0001	0.87	0.65–1.16	0.337
Postoperative reoperation	0.98	0.66–1.45	0.925	--	--	--
Pathological stage AJCC 8th (vs. stage 0)			<0.0001			<0.0001
Stage I	2.21	0.68–7.16	0.187	2.82	0.86–9.27	0.087
Stage II	4.73	1.49–15.00	0.008	4.91	1.53–15.75	0.007
Stage III	9.69	3.08–30.42	<0.0001	8.21	2.57–26.17	<0.0001
Stage IV	31.97	10.23–99.96	<0.0001	20.36	6.29–64.83	<0.0001
Mucinous tumors	1.39	1.11–1.76	0.005	1.11	0.87–1.41	0.409
Signet-ring cells phenotype	2.58	1.48–4.47	0.001	2.24	1.26–3.98	0.006
Extramural invasion	2.64	2.19–3.18	<0.0001	1.41	1.14–1.74	0.001
Lymphovascular invasion	2.68	2.24–3.21	<0.0001	1.31	1.06–1.62	0.013
Perineural invasion	3.29	2.66–4.03	<0.0001	1.44	1.14–1.81	0.002

Abbreviations: R/PFS, recurrence/progression-free survival; HR, hazard ratio; CI, confidence intervals; BMI, body mass index; AJCC, American Joint Committee on Cancer.

**Table 6 cancers-14-06239-t006:** Cox proportional hazard regression analysis on RFS of stage I patients.

	Univariable Analysis			Multivariable Analysis		
Variable	HR	95% CI	*p*	HR	95% CI	*p*
Age of onset (vs. LOCRC)	2.68	1.07–6.72	0.035	2.49	1.19–5.23	0.016
Gender (vs. males)	0.52	0.25–1.07	0.079	--	--	--
Obesity (BMI ≥ 30 Kg/m^2^)	0.91	0.38–2.18	0.828	--	--	--
Smoking (vs. non-smokers)			0.246	--	--	--
Current smokers	1.23	0.49–3.08	0.656	--	--	--
Ex-smokers	1.87	0.89–3.91	0.094	--	--	--
Tumor location (vs. right colon)			0.452	--	--	--
Left colon	0.54	0.18–1.65	0.277	--	--	--
Rectum	0.98	0.44–2.19	0.964	--	--	--
Synchronous tumors	3.08	0.74–12.88	0.123	--	--	--
Family history of CRC	0.32	0.10–1.06	0.063	--	--	--
Postoperative reoperation	0.62	0.13–3.00	0.551	--	--	--
Mucinous tumors	0.84	0.26–2.74	0.775	--	--	--
Extramural invasion	3.17	0.96–10.46	0.057	--	--	--
Lymphovascular invasion	1.10	0.39–3.12	0.852	--	--	--
Perineural invasion	3.01	0.41–22.08	0.277	--	--	--

Abbreviations: R/PFS, recurrence/progression-free survival; HR, hazard ratio; CI, confidence intervals; LOCRC, late-onset colorectal cancer; BMI, body mass index; CRC, colorectal cancer.

## Data Availability

The datasets analyzed in the current study are available from the corresponding author on reasonable request.

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
