# Peer review of "Different Oncologic Outcomes in Early-Onset and Late-Onset Sporadic Colorectal Cancer: A Regression Analysis on 2073 Patients"

_cancers, 2022, doi:10.3390/cancers14246239_

Round 1

Reviewer 1 Report

Thank you for allowing me to review this article. This is a retrospective analysis of a prospective clinical database. Among 2578 patients, one in 5 (505) were excluded. What were the reasons for this?

Neoadjuvant therapy was significantly more frequent in younger patients compared to older patients (38% vs 26%). What were the indications for neo-adjuvant treatment in the 161 young patients when none of them had a rectal location and 144 patients were metastatic at diagnosis?

To the best of our knowledge, 17/423 young patients were operated on as an emergency procedure and 406 as an elective procedure. How can we explain the significant difference in the approach between the youngest and oldest patients, knowing that only 3 out of 4 young patients were operated on laparoscopically and none of them had a rectal tumour? 

130 out of 144 patients were operated on. What was the indication for surgery in these 14 patients in whom no procedure was performed on the metastases?

How do the authors explain the significant increase in the length of hospitalization in the younger patients, whereas the older ones have more comorbidities and the morbidity was significantly higher. Furthermore, younger patients are not as likely to be transferred to a follow-up care facility as older people.  The authors justify their results by a combined liver resection. It would be interesting to know what type of liver resection was performed to explain this prolonged length of stay.

It would be preferable to put the adjuvant treatment paragraph before the survival curves and the information about tumour recurrence

Author Response

Reviewer 1

  1. Thank you for allowing me to review this article. This is a retrospective analysis of a prospective clinical database. Among 2578 patients, one in 5 (505) were excluded. What were the reasons for this?

We would like to thank the reviewer to raise this important point. We reported a brief list of exclusion criteria in the methods section. Since this is a critical point and has been also highlighted by reviewer 2, we decided to extend the eligibility criteria description and report a graphical flowchart in the supplementary material. Briefly, the main exclusion criteria include palliative surgery (e.g. colostomy or ileostomy construction), local excision (by endoscopy or transanal surgery), surgical indication for benign lesions (i.e: adenomatous polyps not endoscopically removable), diagnosis of anal spinocellular cancer, histological diagnosis different from colorectal adenocarcinoma, concomitant diagnosis of inflammatory bowel diseases, known genetic syndromes, and a significant proportion of missing data (with a threshold of 5%) [Page 2, Lines 74-78].

  1. Neoadjuvant therapy was significantly more frequent in younger patients compared to older patients (38% vs 26%). What were the indications for neo-adjuvant treatment in the 161 young patients when none of them had a rectal location and 144 patients were metastatic at diagnosis?

We thank the reviewer for arising this point. 100 Early Onset Rectal Cancers (EORC) underwent neoadjuvant radio-chemotherapy for a locally advanced rectal cancer. 32 EORC were metastatic at diagnosis and underwent both radio-chemotherapy for the rectal cancer and systemic chemotherapy for the metastases. Instead, 29 EORC patients underwent a systemic chemotherapy as they were metastatic at diagnosis and a radio-chemotherapy for their rectal cancer was not indicated (intraperitoneal rectal cancer).  375 late onset rectal cancers (LORC) underwent neoadjuvant radio-chemotherapy for a locally advanced rectal cancer. 54 were metastatic at diagnosis and underwent both radio-chemotherapy for the rectal cancer and systemic chemtherapo for the metastases. Instead, 36 LORC patients underwent a systemic chemotherapy as they were metastatic at diagnosis and a radio-chemotherapy for their rectal cancer was not indicated (intraperitoneal rectal cancer).

We understand that the way data were reported was confusing. We therefore changed the table splitting patients who underwent radio-chemio vs patients who underwent radio-chemo + systemic chemo vs patients who underwent systemic chemo (in this last case we indicated both rectal cancer and colon cancer patients who underwent preoperative systemic chemotherapy). We also changed the text accordingly (Paragraph Demographics and clinical presentation, line 136).

  1. To the best of our knowledge, 17/423 young patients were operated on as an emergency procedure and 406 as an elective procedure. How can we explain the significant difference in the approach between the youngest and oldest patients, knowing that only 3 out of 4 young patients were operated on laparoscopically and none of them had a rectal tumour? More advanced in EOCRC and this explains the higher number of emergency

Thanks to the reviewer for making the point. EOCRC were diagnosed at a more advanced stage and often were very symptomatic at the time of diagnosis. Sometimes the diagnosis was made in the ER at the time they were operated for the primary tumor. The main reason for the emergency surgery was bowel obstruction.  This explains the significant difference in both the operation setting (emergency vs elective) and in the surgical approach (open vs laparoscopic). None of extraperitoneal rectal cancer patients operated in emergency underwent the resection of the primary tumor in emergency. They rather underwent a colostomy to manage colonic obstruction in order to complete staging or to begin neoadjuvant radio-chemotherapy.

  1. 130 out of 144 patients were operated on. What was the indication for surgery in these 14 patients in whom no procedure was performed on the metastases?

Thanks to the reviewer for the question. These 14 patients underwent a disease progression and therefore they did not undergo surgery for the metastases as further systemic chemotherapy was indicated.

  1. How do the authors explain the significant increase in the length of hospitalization in the younger patients, whereas the older ones have more comorbidities and the morbidity was significantly higher. Furthermore, younger patients are not as likely to be transferred to a follow-up care facility as older people.  The authors justify their results by a combined liver resection. It would be interesting to know what type of liver resection was performed to explain this prolonged length of stay.

Thanks to the reviewer for arising this point. Since the first time we thought that this was an interesting result which could have different interpretations, all of them contributing in different proportions to the result: 1) more EOCRC underwent surgery in an emergency setting (with a consequent slower recovery and length of stay); 2) more EOCRC underwent surgery by laparotomy (due both to the operation in an emergency setting and combined complex liver resections  with a consequent slower recovery and length of stay); 3) we also noticed -but this is only a feeling by the authors- an increased fear and anxiety in EOCRC which may have contributed to the longer length of stay. Of course, this is only a feeling by the authors which, however, may be worth to be further investigated 4) combined surgery on CRC and metastases.

To answer the reviewer’s question about the type of liver resection, most of EOCRC underwent a major hepatectomy (i.e.: resection of 4 or more liver segments).

  1. It would be preferable to put the adjuvant treatment paragraph before the survival curves and the information about tumour recurrence

We thank the reviewer for the suggestions. We made the suggested changes.

Reviewer 2 Report

This paper purposed to investigate whether the age of onset may impact on disease recurrence/progression in sporadic colon and rectal cancers analyzed together. I do have some comments as listed below in the order noted.

Comment 1:

What is the novelty of this study although several “Different oncologic outcomes in early-onset and late-onset” studies have been proposed earlier?

Comment 2:

In Introduction, the authors claim that the incidence of early-onset colorectal cancer (EOCRC -age of onset < 49 years) has progressively risen worldwide. The authors should cite the references Gao XH et al., Int J Surg 2022;104:106780; Collaborative R et al., JAMA Surg 2021;156(9):865-874; Himbert C et al., Cancers (Basel) 2021;13(15):3817; and Alyabsi M et al., Front Oncol 2021;11:730689, which were to explore the effect of early age of onset in a surgical population of CRC patients.

Comment 3:

Please add a paragraph about the contribution/significance of this article in a bulleted form at the end part of the Introduction section.

Comment 4:

The quality of the data set is very important, especially for a tertiary referral center ambidirectional parallel-cohort study. For this reason, please clarify the inclusion criteria and exclusion criteria of sample collection in the subsection of Study Design and Patients’ Selection and please also provide a flowchart immediately at the subsection.

Comment 5:

Authors should describe more details of how to choose the variables used in his study.

Comment 6:

Please also add the strengths or significances of the present study in the Discussion section.

Author Response

Reviewer 2

This paper purposed to investigate whether the age of onset may impact on disease recurrence/progression in sporadic colon and rectal cancers analyzed together. I do have some comments as listed below in the order noted.

  1. What is the novelty of this study although several “Different oncologic outcomes in early-onset and late-onset” studies have been proposed earlier?

Thanks to the reviewer for the question. The novelty of this study relies in the finding that early age of onset is an independent risk factor for a worse prognosis, particularly at earlier stages. Our group was the first to report this finding in a comparative study between early- and late- onset rectal cancer and this larger study on colorectal cancer strongly confirms our previous results. Additionally, our study gives a contribution to the still contrasting literature about oncologic outcomes in early onset colorectal cancer patients. In our study -differently from other reports - we did not find differences in the anatomopathologic characteristics between early and late-onset CRC. Hence, other -and still unknown- biological and molecular features could be responsible for the worse prognosis at stage I. We think that our results may contribute to the still scarce literature on the topic and may open the way to further prospective studies aiming to better understand the biology of early-onset colorectal cancer. This could have also a clinical impact to better define management, treatment and follow-up of early-onset colorectal cancer patients. Additionally, the large amount of data, the strict cut-off age and the only inclusion of sporadic cases give an additional value to our results.

  1. In Introduction, the authors claim that the incidence of early-onset colorectal cancer (EOCRC -age of onset < 49 years) has progressively risen worldwide. The authors should cite the references Gao XH et al., Int J Surg2022;104:106780; Collaborative R et al., JAMA Surg 2021;156(9):865-874; Himbert C et al., Cancers (Basel) 2021;13(15):3817; and Alyabsi M et al., Front Oncol 2021;11:730689, which were to explore the effect of early age of onset in a surgical population of CRC patients.

We would like to thank the reviewer for the valuable suggestions. We included the suggested reports within the references of the manuscript [Page 2, Line 42].

  1. Please add a paragraph about the contribution/significance of this article in a bulleted form at the end part of the Introduction section.

Many thanks to the reviewer for the suggestion. We added a paragraph about the contribution/significance of the article at the end of the introduction section.

  1. The quality of the data set is very important, especially for a tertiary referral center ambidirectional parallel-cohort study. For this reason, please clarify the inclusion criteria and exclusion criteria of sample collection in the subsection of Study Design and Patients’ Selection and please also provide a flowchart immediately at the subsection.

We would like to thank the reviewer for this valuable suggestion. We detailed the inclusion and exclusion criteria within the methods section and added an inclusion flowchart in the supplementary material [Page 2, Lines 74-78].

  1. Authors should describe more details of how to choose the variables used in his study. 

Thanks to the reviewer for the suggestion. All relevant demographic, clinical, radiological, endoscopic, surgical, pathological, and oncological data are collected prospectively in our systematic registry. For the present manuscript’s purpose, only a selected set of variables were reported in the table. We decided to provide an extensive list of all variables collected in the supplementary materials (please see Table S1) [Page 2, Lines 70-71; Page 2, Lines 87-89].

  1. Please also add the strengths or significances of the present study in the Discussion section.

Thanks to the reviewer for the suggestion. We added the strengths of the study in the discussion section.

Round 2

Reviewer 1 Report

The authors have answered the questions posed point by point and have adequately corrected the manuscript.